# Locally Advanced Cervical Cancer: Neoadjuvant Treatment versus Standard Radio-Chemotherapy—An Updated Meta-Analysis

**DOI:** 10.3390/cancers16142542

**Published:** 2024-07-15

**Authors:** Carlo Ronsini, Maria Cristina Solazzo, Eleonora Braca, Giada Andreoli, Maria Giovanna Vastarella, Stefano Cianci, Vito Andrea Capozzi, Marco Torella, Luigi Cobellis, Pasquale De Franciscis

**Affiliations:** 1Department of Woman, Child and General and Specialized Surgery, University of Campania “Luigi Vanvitelli”, 80138 Naples, Italy; mariacristinasolazzo@gmail.com (M.C.S.); eleonorabraca9@gmail.com (E.B.); giadaandreoli@gmail.com (G.A.); mariagiovanna.vastarella@unicam.pania.it (M.G.V.); marco.torella@unicampania.it (M.T.); luigi.cobellis@unicampania.it (L.C.); pasquale.defranciscis@unicampania.it (P.D.F.); 2Department of Woman and Child Health, IRCCS, Fondazione Policlinico Gemelli, 00136 Rome, Italy; stefano.cianci@policlinicogemelli.it (S.C.); vitoandreacapozzi@unipr.it (V.A.C.)

**Keywords:** locally advanced cervical cancer, neoadjuvant treatments, radio chemotherapy, oncological outcome

## Abstract

**Simple Summary:**

Cervical cancer is among the most common cancers in women and the fourth leading cause of cancer-related death in females. In developing countries, 6% of diagnoses occur in the advanced stages of the disease. Locally advanced cervical cancer (LACC) is defined as stage IB2-IVA, according to the FIGO 2009 classification. Chemoradiotherapy (CCRT) is the preferred treatment, but 30% of patients relapse after treatment and may suffer from late toxicity. Neoadjuvant chemotherapy (NACT) followed by radical surgery is a common but controversial option, as it can delay curative treatments. Adjuvant surgery after CCRT has shown promising results, with studies indicating potential benefits in disease control and survival rates. However, guidelines mainly suggest exclusive chemoradiotherapy, though a proportion of patients might need additional treatment with surgery. This study compares the oncological outcomes of LACC patients undergoing standard treatments versus those with adjuvant surgery post-CCRT or NACT. Initial findings suggest that adjuvant surgery could improve local control and overall survival in specific patient groups, though further research is needed to confirm these results.

**Abstract:**

Background: The treatment of choice for patients with locally advanced cervical cancer (LACC) is definitive concurrent radio chemotherapy which consists of external beam radiotherapy (EBRT) and concurrent platinum-based chemotherapy (CCRT), with the possible addition of brachytherapy (BT). However, the benefits of adjuvant surgery after neoadjuvant treatments remain a debated issue and a still open question in the literature. This meta-analysis aims to provide an updated view on the controversial topic, focusing on comparing surgery after any adjuvant treatment and standard treatment. Methods: Following the recommendations in the preferred reporting items for systematic reviews and meta-analyses (PRISMA) statement, the PubMed and Embase databases were systematically searched in April 2023 for early publications. No limitations on the country were applied. Only English articles were considered. The comparative studies containing data about disease-free survival (DFS) and/or overall survival (OS) were included in the meta-analysis. Results: The CCRT + surgery group showed a significantly better DFS than CCRT (RR 0.69 [95% CI 0.58–0.81] *p* < 0.01) and a better OS (RR 0.70 [95% CI 0.55–0.89] *p* < 0.01). Nine studies comparing neoadjuvant chemotherapy (NACT) plus surgery and CCRT were also enrolled. The NACT + surgery group showed a significantly better DFS than CCRT (RR 0.66 [95% CI 0.45–0.97] *p* < 0.01) and a better OS (RR 0.56 [95% CI 0.38–0.83] *p* < 0.01). In the sub-analysis of three randomized control trials, the surgery group documented a non-significantly better DFS and OS than CCRT (OR 1.10 [95% CI 0.67–1.80] *p* = 0.72; I^2^ = 69% *p* = 0.72; OR 1.09 [95% CI 0.63–1.91] *p* = 0.75; I^2^ = 13% *p* = 0.32). Conclusion: The results provide updated findings about the efficacy of neoadjuvant treatments, indicating significantly improved DFS and OS in patients undergoing hysterectomy after CCRT or NACT compared with patients undergoing standard treatments.

## 1. Introduction

Cervical cancer remains the fourth most common cancer among women, and it is also the fourth leading cause of cancer-related death in females [1]. It is estimated that 60% of new cervical cancer diagnoses in developing countries are at an advanced stage. Locally advanced cervical cancer (LACC) is defined as a disease in stages ranging from IB2 to IVA, according to the 2009 International Federation of Gynaecology and Obstetrics classification [2]. It represents a particular subset of patients with an estimated 70% overall 5-year survival rate [3]. Several clinical trials have suggested that chemoradiation therapy (CCRT) increases the overall survival (OS) for LACC patients compared to radiotherapy alone [4,5]. Based on these findings, definitive platinum-based chemoradiotherapy and brachytherapy are the preferred treatments [6,7]. Despite the survival advantages compared to radiotherapy alone, more than 30% of patients still experience recurrences after treatment [6]. Also, late toxicity of CCRT treatments may reduce these women’s quality of life [7]. Therefore, several alternative strategies have been tested over time to identify the optimal management of LACC. Neoadjuvant chemotherapy (NACT) followed by radical surgery is still a common option. It has been reported to improve oncological outcomes in selected LACC patients compared to radiotherapy alone, reducing intraoperative spreading risk and postoperative complications [8]. In the era of CCRT, the efficacy of NACT remains controversial, considering potential disadvantages such as the prolongation of curative treatments and disease progression in the case of chemoresistance. Adjuvant surgery after CCRT has also been explored with encouraging results [9]. In addition, variations in chemotherapy type or radicality of surgery within the same strategy make standardization increasingly difficult. Similarly, numerous studies have commented on the risk of comorbidities associated with combining surgical and radiation treatments [10]. Therefore, to date, the treatment suggested by major guidelines is exclusive radio-chemotherapy treatment [11]. However, a proportion of patients do not respond to exclusive treatment and still require supplementation with surgery. Even if, hypothetically, radiation remediation followed by demolition surgery were to guarantee the most radical approach, it is still unclear what its significance is in terms of prognosis and its cost in terms of comorbidities. In this scenario, our work aims to provide a solid overview of the topic, focusing on the oncological outcomes of LACC patients undergoing standard treatments compared to adjuvant surgery after CCRT or NAC.

## 2. Material and Methods

The methods for this study were specified a priori based on the recommendations in the preferred reporting items for systematic reviews and meta-analyses (PRISMA) statement [12]. We registered the review to the PROSPERO site for meta-analysis with protocol CRD42023403900.

### 2.1. Search Method

A systematic search for articles about neoadjuvant treatments and local advanced cervical cancer (LACC) in the PubMed database and Embase database was performed in April 2023. No data limitations were applied. No restrictions of the country were applied. Only fully published studies in English were considered. Search imputes were (“Neoadjuvant Treatments” [Mesh]) AND “Local Advanced Cervical Cancer” [Mesh] for PubMed and Embase databases.

### 2.2. Study Selection

The study selection was done independently by RM and MCS. In the case of discrepancy, CR decided on the inclusion and exclusion criteria. Inclusion criteria were as follows: (1) studies that included patients treated for local advanced cervical cancer FIGO 2009 [2] stage IB2-IVA, with or without nodal involvement, or FIGO 2018 stage IB3-IVA [13]; (2) studies that reported at least one outcome of interest (disease-free survival [DFS] and/or OS, site of recurrence, and major complication of the treatment according to Clavien-Dindo [14]); (3) peer-reviewed articles published originally. Non-original studies, preclinical trials, animal trials, abstract-only publications, and articles in languages other than English were excluded. The studies selected and all reasons for exclusion are mentioned in the preferred reporting items for systematic reviews and meta-analyses (PRISMA) flowchart (Figure 1). All included studies were assessed regarding potential conflicts of interest.

### 2.3. Statistical Analysis

Heterogeneity among the studies was tested using the Chi-square and I-square tests [15]. The risk rate (RR) and 95% confidence intervals (CI) were used for dichotomous variables. Statistical analysis was conducted using fixed-effect models if heterogeneity was low (I^2^ < 50%), or random-effect models if I^2^ > 50%. DFS and OS were used as clinical outcomes. In each study, disease-free survival was defined as the time elapsed between surgery and recurrence or the date of the last follow-up. Overall survival was defined as the time elapsed between surgery and death for cervical cancer or the date of the last follow-up. Review Manager version 5.4.1 (REVman 5.4.1) and IBM Statistical Package for Social Science (IBM SPSS vers 25.0) for MAC were used for statistic calculation. For all performed analyses, a *p*-value < 0.05 was considered significant.

### 2.4. Quality Assessment

Assessment of the quality of the included studies was conducted using the Newcastle–Ottawa scale (NOS) [16]. This assessment scale uses three broad factors (selection, comparability, and exposure), with the scores ranging from 0 (lowest quality) to 8 (best quality). Two authors (CR and MSC) independently rated the study’s quality. Any disagreement was subsequently resolved by discussion or consultation with FF. We used Egger’s regression test to determine the asymmetry of funnel plots.

## 3. Results

### 3.1. Studies’ Characteristics

After the database search, 1001 articles matched the search criteria. After removing records with no full text, duplicates, and wrong study designs (e.g., reviews or non-comparative studies), 24 matched the inclusion criteria and were included in the meta-analysis. Of those, 16 were comparative studies between CCRT and CCRT followed by surgery, and nine compared CCRT with NACT followed by surgery. One study being included, the one by Shanmugam S. et al. [17], showed data from both comparison groups. The countries where the studies were conducted, the publication year range, the studies’ design, the FIGO stage of cervical cancer [2], and the number of participants are summarized in Table 1. The quality of all studies was assessed via NOS [16]. Overall, the publication years ranged from 2007 to 2022, and the follow-up period ranged from 28 to 190 months on average.

Oncological Outcomes: A total of 5946 patients with LACC were included in the review. Of the 24 selected studies, 16 compared CCRT versus CCRT plus surgery. In this group, Hass P. [24] only showed data on DFS, and Albert A. [18] and Cherau E. [40] only showed data on OS. The other 9 studies were about CCRT versus NACT plus surgery and showed data on DFS and OS, except for Ryu H [29] who only presented data on OS. We summarized data about oncological outcomes from each strategy in Table 2.

### 3.2. Treatment Modalities

Therapeutic approaches were highly heterogeneous among analyzed studies. Both NACT and CCRT were platinum-based single-drug or multi-drug combination regimens. The chemotherapy regimens mostly included cisplatin (40 mg/m^2^ weekly, ranging from 20 to 40 mg/m^2^) with a median of six cycles (ranging from 1 to 7). Other chemosensitization schema included weekly administration of 5-fluorouracil or carboplatin (AUC 2) in cases of renal failure. The combination drug regimen included the following: three or two cycles of paclitaxel (175 or 135 mg/m^2^) plus cisplatin (50 or 75 mg/m^2^), 70 or 20 mg/m^2^ of cisplatin plus 700 or 1000 mg/m^2^ of 5-fluorouracil, and two cycles of cisplatin (40 mg/m^2^) plus topotecan (2 mg/m^2^). Triple combination schemes included cisplatin + bleomycin + vincristine or paclitaxel + cisplatin + ifosfamide (5 g/m^2^).

External beam radiation doses ranged from 40 to 80 Gy. In all the reported data, this was followed by intracavitary uterovaginal brachytherapy (20 to 50 Gy) or a parametrial or nodal boost. Adjuvant surgery consisted of a simple hysterectomy or radical hysterectomy, with or without lymph node staging.

### 3.3. CCRT vs. CCRT plus Hysterectomy

All the data about surgical modalities are reported in Table 3.

### 3.4. CCRT vs. NACT plus Surgery

Data about chemotherapeutic modalities are reported in Table 4.

### 3.5. Complete Response Rate, Toxicity and Type of Recurrence

Six studies reported data about the complete response rate after exclusive CCRT, which ranged from 34.2% to 85%. Also, 12 studies reported data about the complete response rate after any neoadjuvant treatment. Overall, rates ranged from 12.8% to 85.9%. Stratification within this range shows an incidence of complete response between 12.8% and 35.8% for chemotherapy-based neoadjuvant treatment and between 27.7% and 85.9% for radiotherapy-based neoadjuvant treatment. Data about the complete response rate are reported in Table 5.

Eight studies also focused on data about surgical complications after neoadjuvant treatment. Data about these complications are reported in Table 6.

Additionally, we reported data about the recurrence rate for each group of patients, which are summarized in Table 7.

We also performed stratification for the site of recurrence, dividing it into local recurrence, systemic recurrence, and both. A total of 13 studies reported these data, and are summarized in Table 8.

Overall, the total local recurrence rate was 10.6% (128/1205) in the surgical group and 13% (193/1479) in the group treated with exclusive CCRT (*p* = NS). Two of the cited studies missed data about the risk of recurrence [17,38].

In a sub-analysis of all the studies with CCRT followed by simple hysterectomy as adjuvant surgical treatment, the total local recurrence rate was 6% (5/81) in the surgical group and 16% (20/127) in the group without adjuvant surgery (*p* < 0.05).

Additionally, in radical hysterectomy approaches, the total local recurrence rate was 4.5% (9/198) in the surgical group and 9% (24/265) in the group treated with exclusive CCRT (*p* = NS).

Only three authors [7,25,32] using neoadjuvant chemotherapy approaches have analyzed in detail the site of relapses. Overall, the total local recurrence rate was 13.7% (63/458) in the surgical group and 11.3% (55/485) in the group treated with exclusive CCRT (*p* = NS).

### 3.6. Meta-Analysis

The 14 studies comparing CCRT and CCRT plus surgery were enrolled in the first meta-analysis. A total of 2544 patients were analyzed. A comparison was made between 1194 patients in the CCRT + surgery arm and 1350 patients who underwent exclusive chemoradiation therapy, exploring DFS outcomes. The fixed-effects model was applied because of low heterogeneity (I^2^ = 38%; *p* = 0.08). The CCRT + surgery group showed a significantly better DFS than CCRT (RR 0.69 [95% CI 0.58–0.81] *p* < 0.01). Nine studies comparing NACT plus surgery and CCRT were also enrolled. Eight of those explored DFS outcomes. A total of 1600 patients were analyzed. A comparison was made between 825 patients in the NACT + surgery arm and 775 patients in the CCRT arm. The random-effects model was applied because of high heterogeneity (I^2^ = 85%; *p* < 0.001). The NACT + surgery group showed a significantly better DFS than CCRT (RR 0.66 [95% CI 0.45–0.97] *p* < 0.01).

Overall, 22 studies compared DFS in patients undergoing neoadjuvant treatments (NADJ) followed by surgery and standard CCRT. A total of 4144 were analyzed. We compared 2019 women in the NADJ arm with 2125 in the CCRT group. Because of high heterogeneity (I^2^ = 70%; *p* < 0.001), a random-effects model was applied. The NADJ + surgery group showed a significantly better DFS than CCRT alone (RR 0.69 [95% CI 0.58–0.83] *p* < 0.01), as shown in Figure 2.

Data on the OS were presented in 15 out of the 16 selected studies. A total of 5694 patients were analyzed. A comparison was made between 1272 patients in the CCRT+ surgery arm and 2784 patients who underwent exclusive chemoradiation therapy. Because of high heterogeneity (I^2^ = 64%; *p* < 0.001), a random-effects model was applied. The CCRT + surgery group showed a significantly better OS than CCRT (RR 0.70 [95% CI 0.55–0.89 *p* < 0.01). Nine studies comparing OS in NACT plus surgery patients with standard CCRT were included, totaling 1732 patients. The NACT + surgery group showed a significantly better OS than CCRT (RR 0.56 [95% CI 0.38–0.83] *p* < 0.01). Because of high heterogeneity (I^2^ = 64%; *p* < 0.01), the random-effects model was applied.

Overall, a total of 5696 patients were analyzed, 2178 in the NADJ arm and 3516 in the CCRT group. Because of high heterogeneity (I^2^ = 64%; *p* < 0.01), a random-effects model was applied. The NADJ + surgery group showed a significantly better OS than CCRT alone (RR 0.66 [95% CI 0.54–0.80] *p* < 0.01), as shown in Figure 3.

We performed a sub-analysis of the published DFS and OS results of three randomized control trials, 175 patients for the CCRT + surgery group and 163 for the CCRT group. In this analysis, the surgery group documented a non-significantly better DFS and OS than CCRT (OR 1.10 [95% CI 0.67–1.80] *p* = 0.72; I^2^ = 69% *p* = 0.72; OR 1.09 [95% CI 0.63–1.91] *p* = 0.75; I^2^ = 13% *p* = 0.32), as shown in Figure 4 and Figure 5.

## 4. Discussion

The utility of completion surgery after neoadjuvant treatments remains a controversial alternative for women with LACC. According to the International Federation of Gynecology and Obstetrics [11,41], the treatment of choice for these patients is definitive platinum-based chemoradiotherapy (CCRT) with or without further interstitial brachytherapy [42,43]. In particular, Keys et al. [4] reported significant differences (*p* < 0.0001) in terms of DFS and OS in the combined therapy group at 4 years. These findings were also corroborated by a meta-analysis [44] that recorded a significant benefit of chemoradiation on both local (odds ratio 0.61, *p* < 0.0001) and distant recurrence (0.57, *p* < 0.0001) in the management of all stages of cervix cancer. More recently, Datta N. et al. [45] conducted a systematic review and meta-analysis to evaluate the efficacy of CCRT over radiotherapy alone, predominantly in LACC. Authors remarked that CCRT significantly improves outcomes in LACC, 10.2% for complete response (*p* = 0.027), 8.4% for local recurrence control (*p* < 0.001), and 7.5% for OS (*p* < 0.001). The data highlight that CCRT is an appropriate option in this subset of patients, but the 5-year OS rate remains around 70% [9]. In this scenario, new treatment modalities have been investigated to eventually improve the oncological outcomes of LACC patients. Among the several therapeutic modalities described, we focused on the role of adjuvant radical hysterectomy after CCRT or NACT compared to standard CCRT. However, a surgical completion should base its rationale on the natural evolution of this pathology. In fact, of the studies reported, the best success rates are still associated with parametrial surgery compared to hysterectomy alone. The Gratz School of Medicine first described the parametrial spread of cervical carcinoma in histological macro-sections. After all, LACCs themselves, while unified within the various papers, may represent three different clinical realities: tumors with a large diameter but limited to the cervix (IB3 FIGO 2018); tumors that, regardless of size, have encroached parameters; tumors that, regardless of locoregional dissemination, have metastasized to the lymph node level. In our opinion, the same treatment principles do not apply to these three realities. Radiation therapy turns to local remediation, with a focus on the macroscopic boundaries of the tumor and minor irradiation of neighboring areas. Surgery, in contrast, performs a cursory exeresis of anatomical districts susceptible to conization by the neoplasm. Consequently, various risk factors should be integrated into the algorithm of therapeutic choice, as they increase the possibility of disseminating the disease to areas other than the starting one (such as grading, histotype, and invasion of lymphovascular spaces) [46,47]. Furthermore, two considerations are necessary. CCRT does not have a 100% success rate and consequently, a proportion of patients will still need to undergo salvage surgical eradication. Secondly, patients whose disease has already spread beyond the cervix can more easily accelerate its spread to other structures. Conversely, all patients undergoing surgery should undergo lymph node staging (the role of the sentinel node seems to be controversial in this scenario) [48].

From this point of view, one out of the three randomized clinical trials conducted on the subject is particularly interesting. Morice et al. [28] treated all the patients with pelvic external radiation therapy with concomitant cisplatin chemotherapy followed by utero-vaginal brachytherapy, offering patients the maximum radiation treatment. Only patients with complete clinical and radiological responses were randomly allocated to radical or extra fascial hysterectomy or no hysterectomy. The results suggested that surgery had no therapeutic impact, but the authors remarked on several limitations of the trial, depending on insufficient accrual. Interestingly, one-third of patients with a complete clinical and radiological response had residual disease in hysterectomy specimens. This finding could suggest the positive impact of surgery treatment. On the other hand, in patients without macroscopic disease, local failure was rare, and no significant differences were detected about the recurrence site in the two arms. Still, this result can also be explained considering that patients at high risk of relapse were excluded from the trial. Also, the other two RTCs did not show superiority of one approach over the other, and this is also mirrored by data from our meta-analysis limited to RCT [17,19]. But, on the other side, authors highlighted that CCRT patients disclosed considerable difficulties in sexual functioning, vaginal functioning, and sexual agony. This may represent a complex issue that impacts an oncological patient’s crucial emotional and physical aspects [49,50], considering that nearly half of all cervical cancer is diagnosed in young women [51]. The results of all the included studies revealed a significantly better DFS and OS in patients treated with CCRT followed by hysterectomy compared to women undergoing standard treatment. By contrast, a previous meta-analysis by Shim et al. [52] indicated no significant difference in OS between the two groups (OR = 1.01; CI = 0.58–1.78; *p* = 0.968) and a better DFS in the surgical group (OR = 0.56; 95% CI = 0.33–0.96; *p* = 0.034). DFS did not differ for the two RCT studies included in a further sub-analysis. The discrepancy can be explained by considering that the authors included only pooled data from two RCT studies and six observational studies. A critical step in identifying the role of surgery in treating LACC may also be verifying recurrence sites. Hypothetically, patients who add surgery to CCRT should have a comparable rate of distant recurrences but a lower rate of local recurrences. On the other hand, one of the concerns related to radical hysterectomy after CCRT is the potential for higher morbidity related to a three-modality approach and the limitation of therapeutic possibilities in case of relapse [53,54]. In our series, data are not fully available. Still, radical surgery seems to be feasible after CCRT [19,23,28,35,38], with a rate of surgical complications within a range comparable to that of radical hysterectomy, carried out as initial management of early-stage cervical cancer [19,55]. Feasibility could be improved with the controversial use of laparoscopy techniques [56,57]. In this context, NACT followed by radical surgery has been proposed as an alternative to standard treatments to achieve several advantages, such as reduction of long-lasting adverse effects linked to radiotherapy, tumor shrinkage, control of local diseases, and increase of resectability [58,59,60]. In our analysis, DFS and OS seem to be better in patients treated with neoadjuvant chemotherapy compared to women undergoing standard CCRT. However, these results have to be interpreted with caution. Several studies, conducted in the era before CCRT, indicated that NACT plus surgery could improve survival compared to conventional radiotherapy (OS: 58.9% vs. 44.5%, *p* = 0.007; DFS: 55.4% vs. 41.3%, *p* = 0.02) [59,61]. Compared to CCRT, the use of NACT is still controversial. Surgery after NACT remains an attractive option to improve disease control and obtain free surgical margins decreasing prognosis risk factors. Nevertheless, disadvantages of NACT could include a delay in curative treatments in the case of a poor chemotherapy response, an increase of cross-resistance, or a prolongation of treatment [25]. Moreover, no study on NACT has explored the possibility of using immunotherapy in PDL-1 patients, representing the latest innovation in therapeutic efficacy.

The present meta-analysis suggested a better DFS and OS in patients treated with neoadjuvant treatment than standard CCRT. Undoubtedly, these results had certain limitations. At first, the results were mostly based on retrospective studies. In addition, three RCTs included in the CCRT plus surgery analysis did not confirm our findings but were numerically insufficient and with strong limitations, as discussed above. Moreover, the complete response after NACT and the presence of residual disease after CCRT represent significant factors that impact the prediction of better survival or local disease progression. Unfortunately, several studies lacked information. This represents one of the major limitations of our study. Another limitation was the lack of information regarding clinical parameters such as age and performance status. It is reasonable to assume that younger patients are less exposed to the comorbidity associated with combined treatments and less prone to relapse. Unfortunately, none of the reported studies performed an age stratification of the sample. The heterogeneity of the radio-chemotherapy schemes used and the different surgical procedures regarding radicality represent another limitation. Finally, although our meta-analysis represents the most updated and comprehensive in the literature, an intrinsic limitation is represented by the impossibility of separating patients with lymph node involvement from those with parametrial involvement based on the published data. As already noted, it would be appropriate to compare these approaches in individual subsets of patients. This is a crucial node in the management of the IIB FIGO stage. Patients without high-risk pathological factors, such as positive nodes and involved surgical margins, could benefit from the advantages of a surgery strategy that avoids the complications of radiotherapy and morbidity of CCRT. However, the strength of this meta-analysis can be found in the rigor of research, providing a significant update compared to previous studies, and contributing to the advancement of optimal treatments for LACC patients.

## 5. Conclusions

Currently, the standard treatment for LACC patients is definitive CCRT, but several controversial treatments still exist in the literature. One of these is represented by the multiplicity of clinical manifestations hidden under the term LACC, which deserves a more personalized treatment. Although the best management strategy remains to be characterized, our work provides updated findings about the efficacy of neoadjuvant treatments, indicating significantly improved DFS and OS in patients undergoing hysterectomy after CCRT or NACT compared with patients undergoing standard treatments. However, given the certain limitations of our results, future controlled clinical trials are required to confirm or disprove the advantages of adjuvant surgery in these patients.

## Figures and Tables

**Figure 1 cancers-16-02542-f001:**
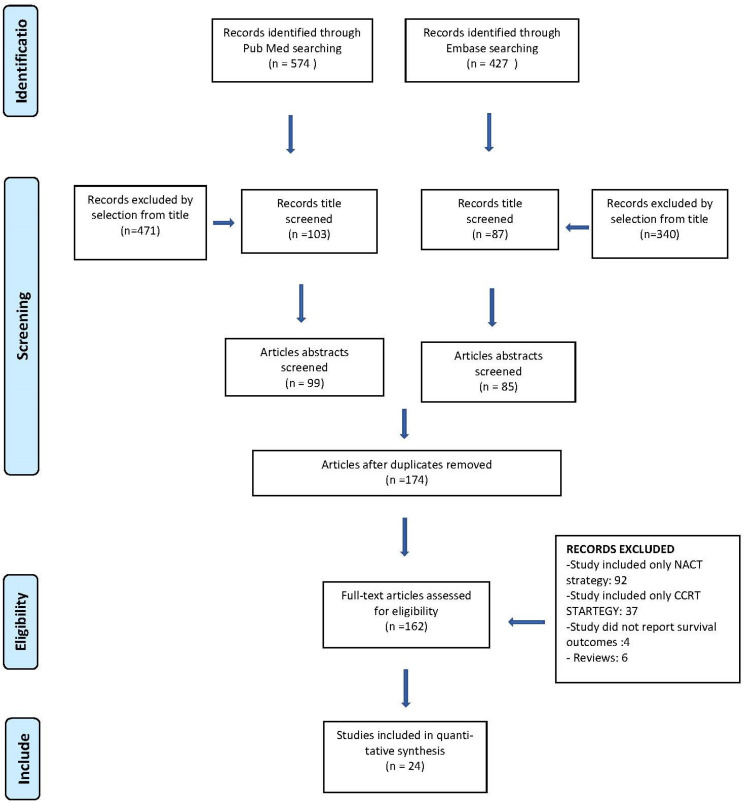
PRISMA flow diagram.

**Figure 2 cancers-16-02542-f002:**
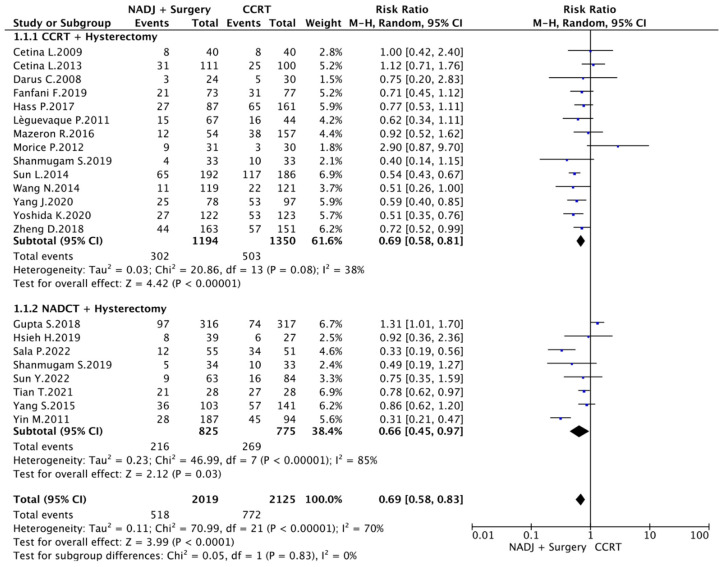
Recurrence Risk [7,17,19,20,22,23,24,25,26,27,28,30,31,32,33,34,35,36,37,38,39].

**Figure 3 cancers-16-02542-f003:**
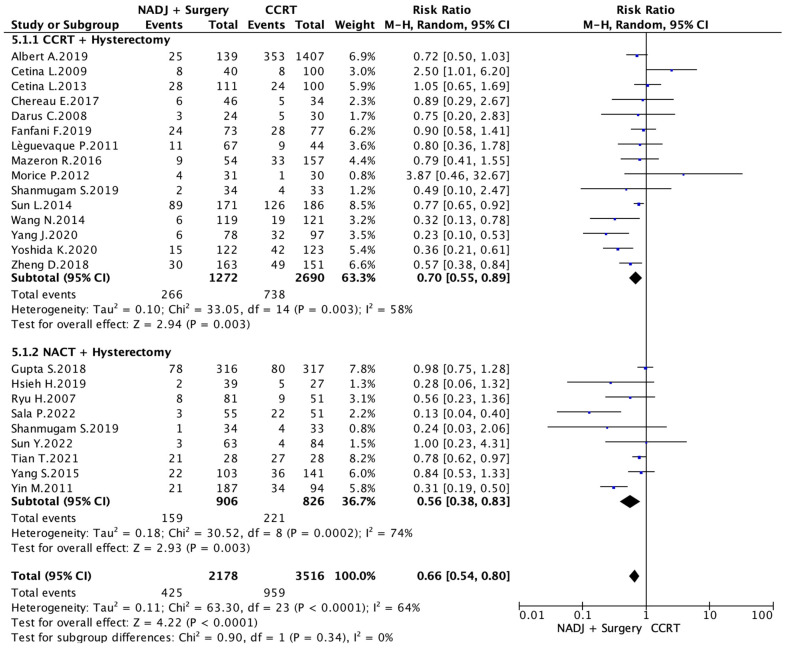
Death Risk [7,17,18,19,20,21,22,23,24,25,26,27,28,29,30,31,32,33,34,35,36,37,38,39].

**Figure 4 cancers-16-02542-f004:**
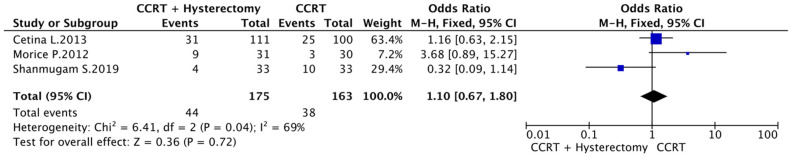
Recurrence Risk RCT [17,20,28].

**Figure 5 cancers-16-02542-f005:**
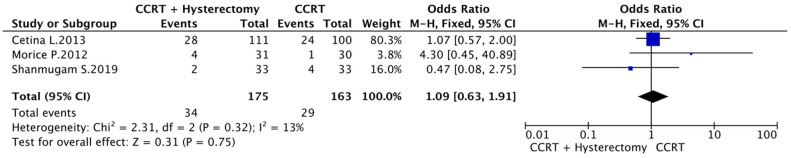
Death Risk RCT [17,20,28].

**Table 1 cancers-16-02542-t001:** Studies included [16,17,18].

Comparative Studies						
Name	Country	Study Design	Study Year	FIGO Stage	N of Partecipant (NADJ/CCRT)	Mean FUP * Months
Albert A., 2019 [18]	USA	Retrospective Case-Control Monocentric Study	2010–2014	IB2 IIA	1546 (139/1407)	33.3
Cetina L., 2013 [19]	Mexico	Prospective Case-Control Monocentric Study	2004–2009	IB2 IIA2 IIB	211 (111/100)	36
Cetina L., 2009 [20]	Mexico	Retrospective matched Control Monocentric Study	1999–2003	IB2 IIA IIB	140 (40/100)	29
Chereau E., 2013 [21]	France	Retrospective Case-Control Monocentric Study	2002–2012	IB2 II	80 (46/34)	30.7
Darus C., 2008 [22]	USA	Retrospective Case-Control Monocentric Study	1994–2004	IB2	54 (24/30)	46.8
Fanfani F., 2019 [23]	Italy	Retrospective Case-Control Multicentric Study	1999–2013	IIIA IIIB	150 (73/77)	40
Gupta S., 2018 [7]	India	Randomized Control Monocentric Trial	2003–2015	IB2 IIA IIB	633 (316/317)	58.5
Hass P., 2017 [24]	Germany	Retrospective Case-Control Multicentric Study	2003–2011	IB2-IVA	248 (87/161)	57
Hsieh H., 2019 [25]	Republic of China	Retrospective Case-Control Monocentric Study	2002–2016	IB2	66 ^ (39/27)	66.2
Lèguevaque P., 2011 [26]	France	Retrospective Case Control Multicentric Study	1989–2006	IB1-IVA	111 (67/44)	-
Mazeron R., 2016 [27]	Russia	Retrospective Case-Control Monocentric Study	2004–2008	IB1 IB2 IIA IIB	211 (54/157)	57,4
Morice P., 2012 [28]	France	Randomized Control Multicentric Trial	2003–2006	IB2 II	61 (31/30)	44
Ryu H., 2007 [29]	Korea	Retrospective Observational Multicentric Study	1995–2005	IB2	132 ^ (81/51)	120
Sala P., 2022 [30]	Italy	Retrospective Case-Control Multicentric Study	2006–2018	IB2-IVA	106 (55/51)	33
Shanmugam S., 2019 [17]	India	Randomized Control Monocentric Trial	2014–2018	IB2 IIA2 IIB	100 (34/33/33)	28
Sun L., 2014 [31]	China	Retrospective Case-Control Monocentric Study	1992–2012	IIB III IVA	378 (192/186)	190
Sun Y., 2022 [32]	China	Retrospective Case-Control Multicentric Study	2013–2019	IB2 IIA2 IIB	147 (63/84)	60
Tian T., 2021 [33]	China	Retrospective Matched Control Monocentric Study	2013–2017	IB2 IIA2 IIB IIIB	56 ^a^ (28/28)	-
Wang N., 2014 [34]	China	Prospective Case-Control Monocentric Study	2004–2011	IIB	240 (119/121)	33
Yang J., 2020 [35]	China	Retrospective Case-Control Monocentric Study	2004–2018	IB IIA IIB III	175 (78/97)	20.5
Yang S., 2015 [36]	China	Retrospective Case-Control Monocentric Study	2007–2009	IIB	244 (103/141)	67
Yin M., 2011 [37]	China	Retrospective Case-Control Monocentric Study	2000–2005	IB2 IIA IIB	281 (187/94)	82.8
Yoshida K., 2020 [38]	Japan	Retrospective Matched Case-Control Monocentric Study	2005–2015	IB2 IIA IIB	245 (122/123)	64.8
Zheng D., 2018 [39]	China	Retrospective Case-Control Monocentric Study	2008–2013	IB2 IIB	314 (163/151)	60

* Follow-up, ^ Sub-analysis of the entire cohort. ^a^: After propensity score matching, NADJ: neoadjuvant treatments; CCRT: radiochemotherapy.

**Table 2 cancers-16-02542-t002:** Oncological Outcomes.

		NADJ						CCRT				
Name	NADJ Treatment	3Y DFS (%)	3Y OS (%)	4.5Y DFS (%)	4.5Y OS (%)	FU	3Y DFS (%)	3Y OS (%)	4.5Y DFS (%)	4.5Y OS (%)	FU	P
Albert A., 2019 [18]	Radiation	-	-	-	82.2%	-	-	-	-	74.9%	33	0.036
Cetina L., 2013 [19]		71.7	74.5	-	-	36	74.8	76.3	-	-	36	0.186 (3YDFS) 0.236
Cetina L., 2009 [20]		-	-	78%	78%	26	-	-	78%	78%	22	NS
Chereau E., 2013 [21]		-	-		86.9%		-	-		85.3%	30.7	NS
Darus C., 2008 [22]		-	-	83,3	87.5	46.8	-	-	86.6	83.3	46.8	0.7 (DFS) 0.8 (OS)
Fanfani F., 2019 [23]		71.23	63.6	-	-	39	59.7	63.6	-	-	41	0.686 (3YDFS) 0.675
Gupta S., 2018 [7]	Chemotherapy	-	-	69.3	75.4	58.5	-	-	76.7	74.7	58.5	0.038 (3YDFS) 0.870
Hass P., 2017 [24]		-	-	68.9	-	53	-	-	59.6	-	53	-
Hsieh H., 2019 [25]		-	-	79.3	94.1	83.5	-	-	79.5	80.1	63.3	0.401 (4.5YDFS) 0.197
Lèguevaque P., 2011 [26]		-	-	77.6	83.5	-	-	-	65.9	79.5	-	0.558 (4.5DFS) 0.296
Mazeron R., 2016 [27]		-	-	77.7	83.3	67.9	-	-	75.8	78.9	42.1	0.926 (4.5DFS) 0.630
Morice P., 2012 [28]		72	86	-	-	44	89	97	-	-	44	NS (3YDFS) NS
Ryu H., 2007 [29]		-	-	-	90	120	-	-	-	83	120	NS
Sala P., 2022 [30]		-	-	77.4	93.8	33	-	-	33.4	56.5	33	<0.001 (4.5DFS) 0.003
Shanmugam S., 2019 [17]		85 88	100 94	-	-	28	70	88	-	-	28	0.571
Sun L., 2014 [31]		83.2	72.2	66.1	47.9	190	54.1	45.9	37.1	32.2	190	<0.005
Sun Y., 2022 [32]		90.5	95.2	86.1	89.9	60	89.3	95.2	80.6	89.9	60	0.849 (3YDFS) 0.816 (3YOS) >0.05 (4.5YDFS7OS)
Tian T., 2021 [33]		-	-	25 ^a^	25 ^b^	-	-	-	4 ^a^	4 ^b^	-	0.00015 (4.5YDFS) 0.00014
Wang N., 2014 [34]		91	94.9	-	-	36	81.8	84.6	-	-	30	0.049 (3YDFS) 0.011
Yang J., 2020 [35]		67.9	92.3	-	-	28	45.4	67	-	-	16	0.002 (3YDFS) 0.002
Yin M., 2011 [37]		-	-	65	78.6	67	-	-	59.4	74.5	67	0.456 (4.5YDFS) 0.637
Yoshida K., 2020 [38]		-	-	85	88.67	82.8	-	-	52	64.37	82.8	<0.0001 (4.5DFS) <0.0001
Zheng D., 2018 [39]		-	-	78.3 ^a^	87.7 ^b^	64.8		-	56.9 ^a^	66.2 ^b^	64.8	0.027 (4.5DFS) 0.017
Yin M., 2011 [37]		77.3	87.1	73.3	81.7	60	67.2	72.8	62.4	67.3	60	0.01 (3–4.5YDFS) 0.001 (3–4.5OS)

NADJ: neoadjuvant treatment, CCRT: exclusive concurrent chemoradiotherapy, S: hysterectomy, CHT: chemotherapy, NACT: neoadjuvant chemotherapy, NS: not significant, ^a^: adjusted 4.5Y-DFS, ^b^: adjusted 4.5Y OS.

**Table 3 cancers-16-02542-t003:** Surgical treatment outcomes.

Name	EBRT (Gy)	BRT Yes/No (Gy)	ADJ Surgery	Pelvic Lymphadenectomy Yes/No
Albert A., 2019 [18]	60	NA	SH	NA
Cetina L., 2013 [19]	50.4	No	RH	Yes
Cetina L., 2009 [20]	50	No ^a^	RH	Yes
Chereau E., 2013 [21]	40	Yes (20)	SH/RH	Yes
Darus C., 2008 [22]	45	Yes (30)	SH	No
Fanfani F., 2019 [23]	50	Yes (30)	RS	Yes
Hass P., 2017 [24]	50.4	No	RS	No ^b^
Lèguevaque P., 2011 [26]	45	Yes (15)	SH/RH	Yes
Mazeron R., 2016 [27]	40–50.4	Yes (20)	SH/RH	No
Morice P., 2012 [28]	45	Yes (15)	SH/RH	Yes ^c^
Shanmugam S., 2019 [17]	50	No	RH	NA
Sun L., 2014 [31]	45–50	Yes (45–55)	SH/RH	No
Wang N., 2014 [34]	40–50	No	RH	Yes
Yang J., 2020 [35]	NA	Yes	SH	No
Yoshida K., 2020 [38]	39.6	No	RH	Yes
Zheng D., 2018 [39]	46–50	Yes (25–30)	RH	Yes

EBRT: external beam radiation therapy, BRT: brachytherapy, ADJ: adjuvant, SH: simple hysterectomy, RH: radical hysterectomy, NA: not available, ^a^: post-operative brachytherapy was performed in cases with one or more high/intermediate-risk factors for recurrence, ^b^: except 28 patients, ^c^: 17 patients underwent para-aortic lymphadenectomy. Six patients underwent complete bilateral pelvic lymphadenectomy, and three had a unilateral pelvic lymphadenectomy.

**Table 4 cancers-16-02542-t004:** Chemotherapeutic treatment outcomes.

Name	Drugs	N of Cycle	ADJ Surgery	Pelvic Lymphadenectomy Yes/No
Gupta S., 2018 [7]	Paclitaxel + Carboplatin	3	RH	Yes
Hsieh H., 2019 [25]	Cisplatin	6	RH	Yes
Ryu H., 2007 [29]	NA	NA	RH	NA
Sala P., 2022 [30]	Platinum based combination	3	RH	Yes
Shanmugam S., 2019 [17]	Paclitaxel + Cisplatin	3	RH	Yes
Sun Y., 2022 [32]	Paclitaxel + Platinum	2	RH	Yes
Tian T., 2021 [33]	Paclitaxel + Cisplatin	2–3	RH	Yes
Yang S., 2015 [36]	cisplatin/nedaplatin/carboplatin + pac-litaxel	1–3	RH	Yes
Yin M., 2011 [37]	Platinum-based com-bination	2–3	RH	Yes

RH: radical hysterectomy; ADJ: adjuvant, NA: not available.

**Table 5 cancers-16-02542-t005:** NADJ/CCRT Complete Response.

Name	NADJ CR (%)	CCRT CR (%)	*p*
Cetina L., 2009 [20]	22 (55%)	34(85%)	0.2
Cetina L., 2013 [19]	62 (72%)	-	-
Darus C., 2008 [22]	13 (67%)	-	
Chereau E., 2013 [21]	27 (12.4%)	-	-
Fanfani F., 2019 [23]	41.5%	38.9%	0.6
Hass P., 2017 [24]	40 (40.6%)	81 (50.3%)	0.5
Hsieh H., 2019 [25]	5 (12.8%)	-	
Shanmugam S., 2019 [17]	24 (35.8%)	24 (72%)	0.001
Sun L., 2014 [31]	165 (85.9%)	-	-
Wang N., 2014 [34]	33 (27.73%)	-	-
Yang J., 2020 [35]	-	28 (35.9%)	
Yoshida K., 2020 [38]	12 (24%)	26(34.2%)	0.2

NADJ: neoadjuvant treatment, CCRT: exclusive concurrent chemoradiotherapy, CR: complete response.

**Table 6 cancers-16-02542-t006:** NADJ Complications rate *.

Name	NADJ Early Complication Rate (%)	NADJ Late Complication Rate (%)	Tot. %
Cetina L., 2013 [19]	0	5 (5.8%)	5.8
Cetina L., 2009 [20]	-	9 (22.5%)	22.5
Chereau E., 2013 [21]	-	-	6.4
Fanfani F., 2019 [23]	5 (6.8%)	3 (4.1%)	10.9
Hass P., 2017 [24]		2 (2.2%)	2.2
Mazeron R., 2016 [27]		11 (20.3%)	20.3
Yang J., 2020 [35]	5 (6.4%)	-	6.4
Yoshida K., 2020 [38]		14 (28%)	28

NADJ: neoadjuvant treatment, * according to Clavien-Dindo ≥ 3.

**Table 7 cancers-16-02542-t007:** Recurrence Rate.

Name	NADJ Recurrence Rate (%), n	Tot.	CCRT Recurrence Rate (%), n	Tot.	*p*
Cetina L., 2009 [20]	8 (20%)	40	8 (20%)	40	1
Cetina L., 2013 [19]	13 (11.7%)	86	15 (15%)	86	0.9
Darus C., 2008 [22]	3 (12.5%)	24	5 (16.6%)	30	ns
Fanfani F., 2019 [23]	21 (28.7)	73	31 (40.2)	77	<0.001
Gupta S., 2018 [7]	59 (18.6%)	316	43 (13.56%)	317	0.08
Hass P., 2017 [24]	27 (31%)	87	65 (40.3%)	161	0.4
Hsieh H., 2019 [25]	7 (17.9%)	39	11 (40.7%)	27	ns
Lèguevaque P., 2011 [26]	15 (2.4%)	67	16 (36.4%)	44	0.01
Mazeron R., 2016 [27]	14 (29.5%)	54	36 (22.9%)	157	0.6
Morice P., 2012 [28]	8 (25.8%)	31	4 (13.3%)	30	0.2
Sun L., 2014 [31]	32 (16.7%)	192	59 (31.7%)	186	0.0006
Sun Y., 2022 [32]	6 (9.5%)	63	12 (14.2%)	84	0.3
Tian T., 2021 [33]	21	28	27	28	0.02
Wang N., 2014 [34]	11 (9.24)	119	22 (18.18)	121	0.06
Yang J., 2020 [35]	16 (33.33%)	78	45 (50.56%)	97	0.025
Yang S., 2015 [36]	59 (57.28%)	103	37 (26.24%)	141	0.3
Yin M., 2011 [37]	28	187	45	94	>0.05
Yoshida K., 2020 [38]	15 (30%)	50	28 (36.8%)	76	0.4
Zheng D., 2018 [39]	48 (29.4%)	163	58 (38.4%)	151	0.009

NADJ: neoadjuvant treatment, CCRT: exclusive concurrent chemoradiotherapy.

**Table 8 cancers-16-02542-t008:** Site of Recurrence.

Name	NADJ Recurrence Site (n)					CCRT Recurrence Site (n)					*p*
	l	s	ls	Tot	FUP *	l	s	ls	Tot	FUP *	
Cetina L., 2009 [20]	6	0	2	8	26	6	1	1	8	22	ns
Cetina L., 2013 [19]	7	6	0	13	36	10	5	0	15	36	0.4 (l) 0.4 (s)
Darus C., 2008 [22]	1	-	-	3	24	1	3	1	5	30	ns
Fanfani F., 2019 [23]	2	3	-	21	39	7	6	-	31	41	<0.002 (l) 0.9 (s)
Gupta S., 2018 [7]	39	11	20	89	58.5	19	22	24	73	58.5	0.01 (l)
Hass P., 2017 [24]	20	7	0	27	58.5	44	21	0	65	58.5	0.5 (l) 0.2 (s)
Hsieh H., 2019 [25]	7	4	0	11	83.5	3	4	0	7	63.3	ns ns
Lèguevaque P., 2011 [26]	6	9	0	15	-	9	7	0	16	-	0.3 (l) 0.3 (s)
Mazeron R., 2016 [27]	1	15	0	16	67.9	11	41	0	52	42.1	0.1 (l) 0.1 (s)
Morice P., 2012 [28]	1	2	5	8	44	2	1	0	3	44	ns
Wang N., 2014 [34]	5	7	1	13	36	8	13	0	22	30	0.5 (l) 0.2 (s)
Yang J., 2020 [35]	4	16	6	26	28	12	19	14	45	16	0.2 (l) 0.2 (s)
Yang S., 2015 [36]	17	13	7	37	67	33	18	8	59	67	0.6 (l) 0.6 (s)
Zheng D., 2018 [39]	12	29	7	48	60	18	32	8	58	60	0.4 (l) 0.5 (s)

NADJ: neoadjuvant treatment, CCRT: exclusive concurrent chemoradiotherapy, l: local recurrence; s: systemic recurrence; ls: local and systemic recurrences, ns: not significant; * follow-up.

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
