# Peer review of "Locally Advanced Cervical Cancer: Neoadjuvant Treatment versus Standard Radio-Chemotherapy—An Updated Meta-Analysis"

_cancers, 2024, doi:10.3390/cancers16142542_

Round 1

Reviewer 1 Report

Comments and Suggestions for Authors

This paper reports different results than generally proposed in literature. However, when the authors limit the analysis to only RCT, difference in PFS or OS disappears. 

results are too long. tables would be enough. this chapter should be re-written and shortened

similarly, the discussion is too long, and lacks a strength/limitation section

Author Response

Dear Reviewer,

Thank You for taking the time to review our manuscript and for your comments. They are crucial and valuable to us in raising the quality standard of our work.

We wanted to inform You that we have made a general revision of the English and grammar. In addition, a specification for Your revisions is below:

“results are too long. tables would be enough. this chapter should be re-written and shortened”

-We followed your advice and changed it according to your observation. We limited results to summarized data.

similarly, the discussion is too long, and lacks a strength/limitation section

- Thank you. We have eliminated repetitive parts and streamlined the discussion

You can find the rewritten and corrected version of the manuscript in the attached file. We highlighted any changes made.

Morover, Grammar correction tools (Grammarly, Inc) were used to improve the quality of English and readability. The technology was used under human oversight and control.

Thank you very much for your advice and comments. We hope we have complied with your requests.

Reviewer 2 Report

Comments and Suggestions for Authors

The authors presented the systematic review study entitled” Locally Advanced Cervical Cancer: Neoadjuvant Treatment versus Standard Radio Chemotherapy: An up-dated Meta-Analysis” based on the recommendations in the Preferred Reporting Items for Systematic Reviews and Meta-Analyses (PRISMA) statement. Tables and figures showed the studies included and comparisons in patient’s survival and outcome. Younger age and less comorbidity could be the major candidate for surgery, and it could lead the better outcome. I suggest age or comorbidity be stated and analyzed for the included studies; especially, the retrospective studies.

Comments on the Quality of English Language

Minor editing of English language required

Author Response

Dear Reviewer,

Thank You for taking the time to review our manuscript and for your comments. They are crucial and valuable to us in raising the quality standard of our work.

We wanted to inform You that we have made a general revision of the English and grammar. In addition, a specification for Your revisions is below:

“The authors presented the systematic review study entitled” Locally Advanced Cervical Cancer: Neoadjuvant Treatment versus Standard Radio Chemotherapy: An up-dated Meta-Analysis” based on the recommendations in the Preferred Reporting Items for Systematic Reviews and Meta-Analyses (PRISMA) statement. Tables and figures showed the studies included and comparisons in patient’s survival and outcome. Younger age and less comorbidity could be the major candidate for surgery, and it could lead the better outcome. I suggest age or comorbidity be stated and analyzed for the included studies; especially, the retrospective studies.”

- We agree that this observation would have an important clinical impact. Unfortunately, none of the studies reported in the meta-analysis reported detailed clinical information about patients who relapsed, making it impossible to follow up this observation. Therefore, we have specified in the limitations of this study the absence of this information.

You can find the rewritten and corrected version of the manuscript in the attached file. We highlighted any changes made.

Morover, Grammar correction tools (Grammarly, Inc) were used to improve the quality of English and readability. The technology was used under human oversight and control.

Thank you very much for your advice and comments. We hope we have complied with your requests.

Reviewer 3 Report

Comments and Suggestions for Authors

The study by Carlo Ronsini and colleagues systematically reviewed the outcomes of patients who underwent either concurrent platinum-based chemotherapy and radiation therapy (CCRT) or neoadjuvant chemotherapy (NACT) followed by surgery for locally advanced cervical cancer (LACC). This research highlights the critical role of surgery in the treatment regimen after CCRT or NACT. The experimental design and statistical analyses employed are sound, guaranteeing the reliability and validity of the results. The methodology is thorough, with well-defined criteria for patient selection, treatment protocols, and the evaluation of response and survival rates. Despite a modest level of novelty in the manuscript, the study offers a robust foundation of evidence for clinical decision-making. Thus I recommend that this manuscript can be accepted with minor revise.

Furthermore, the inclusion of or discussion on the potential benefits of immunotherapy, such as PD-1 antibody in combination with NACT before surgery, could be an area worth exploring.

Author Response

Dear Reviewer,

Thank You for taking the time to review our manuscript and for your comments. They are crucial and valuable to us in raising the quality standard of our work.

We wanted to inform You that we have made a general revision of the English and grammar. In addition, a specification for Your revisions is below:

“Furthermore, the inclusion of or discussion on the potential benefits of immunotherapy, such as PD-1 antibody in combination with NACT before surgery, could be an area worth exploring”

- We agree that this observation will enhance the clinical impact of this work. Unfortunately, no studies on NACHT reported immunotherapy-based schemes. However, in the discussion we reported how immunotherapy could be an additional tool to optimise the complete response rate to treatment.

You can find the rewritten and corrected version of the manuscript in the attached file. We highlighted any changes made.

Morover, Grammar correction tools (Grammarly, Inc) were used to improve the quality of English and readability. The technology was used under human oversight and control.

Thank you very much for your advice and comments. We hope we have complied with your requests.

Round 2

Reviewer 1 Report

Comments and Suggestions for Authors

OK for publication